# Visfatin Facilitates VEGF-D-Induced Lymphangiogenesis through Activating HIF-1α and Suppressing miR-2277-3p in Human Chondrosarcoma

**DOI:** 10.3390/ijms25105142

**Published:** 2024-05-09

**Authors:** Chang-Yu Song, Shang-Lin Hsieh, Shang-Yu Yang, Chih-Yang Lin, Shih-Wei Wang, Chun-Hao Tsai, Yuan-Shun Lo, Yi-Chin Fong, Chih-Hsin Tang

**Affiliations:** 1Graduate Institute of Biomedical Science, China Medical University, Taichung 40402, Taiwan; zxcb983513@gmail.com (C.-Y.S.); neosolomon@msn.com (S.-L.H.); 2Minimally Invasive Spine and Joint Center, Buddhist Tzu Chi General Hospital Taichung Branch, Taichung 42721, Taiwan; 3Department of Medical Laboratory Science and Biotechnology, Asia University, Taichung 41354, Taiwan; henry879019@asia.edu.tw; 4Translational Medicine Center, Shin-Kong Wu Ho-Su Memorial Hospital, Taipei 11104, Taiwan; p123400@hotmail.com; 5Institute of Biomedical Sciences, Mackay Medical College, New Taipei City 25245, Taiwan; shihwei@mmc.edu.tw; 6Department of Medicine, Mackay Medical College, New Taipei City 25245, Taiwan; 7Department of Sports Medicine, College of Health Care, China Medical University, Taichung 40432, Taiwan; ritsai8615@gmail.com (C.-H.T.); yichin.fong@msa.hinet.net (Y.-C.F.); 8Department of Orthopedic Surgery, China Medical University Hospital, Taichung 40432, Taiwan; yuanshunlo@gmail.com; 9Department of Orthopedic Surgery, China Medical University Beigang Hospital, Yunlin 65101, Taiwan; 10Graduate Institute of Precision Engineering, National Chung Hsing University, Taichung 40227, Taiwan; 11Department of Pharmacology, School of Medicine, China Medical University, Taichung 40402, Taiwan; 12Chinese Medicine Research Center, China Medical University, Taichung 40402, Taiwan; 13Department of Medical Research, China Medical University Hsinchu Hospital, Hsinchu 30205, Taiwan

**Keywords:** chondrosarcoma, lymphangiogenesis, VEGF-D, HIF1α, miR-2277-3p

## Abstract

Chondrosarcoma is a malignant bone tumor that arises from abnormalities in cartilaginous tissue and is associated with lung metastases. Lymphangiogenesis plays an essential role in cancer metastasis. Visfatin is an adipokine reported to enhance tumor metastasis, but its relationship with VEGF-D generation and lymphangiogenesis in chondrosarcoma remains undetermined. Our results from clinical samples reveal that VEGF-D levels are markedly higher in chondrosarcoma patients than in normal individuals. Visfatin stimulation promotes VEGF-D-dependent lymphatic endothelial cell lymphangiogenesis. We also found that visfatin induces VEGF-D production by activating HIF-1α and reducing miR-2277-3p generation through the Raf/MEK/ERK signaling cascade. Importantly, visfatin controls chondrosarcoma-related lymphangiogenesis in vivo. Therefore, visfatin is a promising target in the treatment of chondrosarcoma lymphangiogenesis.

## 1. Introduction

A malignant bone tumor called chondrosarcoma develops from irregularities in cartilaginous tissue. It is the second most usual primary malignancy among bone malignancies, accounting for around one-third of all cases [1]. Chondrosarcoma typically occurs near axial bones such as the pelvis and scapula and primarily affects individuals between the ages of 40 and 60 [2]. It also frequently occurs at the proximal extremities of the limbs, such as the proximal humerus and proximal femur [3]. The main treatment for chondrosarcoma at the moment is surgical excision, with survival rates varying depending on the tumor grade [3]. High-grade chondrosarcoma has a higher chance of distant metastasis, resulting in a worse prognosis [4,5]. There is an important require to develop a novel targeted strategy because there are currently no effective treatments for metastatic or unresectable chondrosarcoma.

The primary cause of cancer-related deaths worldwide is metastasis [6,7]. In the majority of human malignancies, the initial step involves metastatic spread to sentinel lymph nodes [8,9]. It has been proposed that there is a link between tumor lymphangiogenesis and lymph node metastases in various types of human cancer [10]. Tumor-secreted lymphangiogenic factors can stimulate the formation of lymphatic vessels. VEGF-D is a secreted glycoprotein that promotes the growth and remodeling of both lymphatic and blood vessels by activating endothelial VEGF receptors and acting as a mitogen for endothelial cells. Lymphatic endothelial cells (LECs) predominantly express VEGF-D, which primarily exerts its functions through VEGFR-3 [11]. During the process of lymphangiogenesis, LECs are reported to proliferate, survive, migrate, and form tubes due to the interaction between VEGF-D and VEGFR-3 [12]. Furthermore, lymphatic metastasis and tumor-associated lymphangiogenesis are reliant on the release of VEGF-D by tumor cells [13].

An expanding body of research suggests a connection between cancer and obesity [14]. This link is particularly evident in cancers such as esophageal, pancreatic, and colorectal cancers [14,15]. Furthermore, obesity can alter the cancer’s microenvironment, accelerating its progression [16]. Visceral fat contains substantial amounts of visfatin, an adipokine that promotes inflammation [17]. Visfatin is known to regulate several cellular functions in mammalian cells, including cellular proliferation, migration, differentiation, and apoptosis [17]. It is not surprising that individuals with various malignancies exhibit significantly higher visfatin levels compared to people without cancer [18]. Visfatin expression is recognized as vital in various tumor-associated activities, including survival, angiogenesis, metastasis, and treatment resistance. Reports have demonstrated that visfatin promotes chondrosarcoma angiogenesis and metastasis [19,20,21]. However, the precise role of visfatin in human chondrosarcoma lymphangiogenesis remains largely unknown. In our current investigation, we discovered that visfatin stimulates the Raf/MEK/ERK pathway and activates HIF-1α as well as suppresses the expression of miR-2277-3p, thereby promoting VEGF-D-mediated lymphangiogenesis in chondrosarcoma. Lastly, we found that inhibiting visfatin suppresses in vivo chondrosarcoma lymphangiogenesis. This discovery positions visfatin as a potential target for the development of therapies targeting chondrosarcoma lymphangiogenesis.

## 2. Results

### 2.1. Positive Correlation of Visfatin and VEGF-D in Human Chondrosarcoma Patients

Given that VEGF-D is known to control lymphangiogenesis in several types of cancer cells [22,23], the levels of VEGF-D in human chondrosarcoma have remained largely unexplored. To address this, we initially examined VEGF-D levels in chondrosarcoma patients. Our analysis of data from IHC staining revealed higher levels of VEGF-D expression in patients with higher-grade chondrosarcoma compared to those with lower-grade disease (Figure 1A,B). We also stained for the lymphatic vessel marker (LYVE-1) and found that human chondrosarcoma expressed a higher level of LYVE-1 than normal cartilage (Appendix A). As previously established, visfatin plays a role in enhancing angiogenesis and metastasis in human chondrosarcoma [19,20]. Visfatin levels are confirmed to be higher in chondrosarcoma patients than in normal cartilage (Appendix A). Notably, our analysis indicated a positive correlation between visfatin and VEGF-D intensity in chondrosarcoma tissues (Figure 1C,D). This finding suggests that the levels of visfatin and VEGF-D are associated with the development of chondrosarcoma.

### 2.2. Exogenous Visfatin Enhances VEGF-D Production and Lymphangiogenesis in Human Chondrosarcomas

We proceeded to examine the impact of visfatin on VEGF-D generation and lymphangiogenesis in chondrosarcoma cell lines. To do this, we directly treated human chondrosarcoma cell lines (JJ012 and SW1353) with visfatin and examined its effect on VEGF-D expression. Our analysis revealed that visfatin facilitates the expression of VEGF-D mRNA and the production of VEGF-D protein in a concentration-dependent manner, as assessed by qPCR, Western Blot, and ELISA assays (Figure 2A–C). During the process of tumor-induced lymphangiogenesis, lymphatic endothelial cells are required to migrate and form tube-like structures to create new lymph vessels. Thus, we explored whether visfatin-induced VEGF-D expression led to lymphangiogenesis. Conditioned medium (CM) obtained from JJ012 and SW1353 cells, treated with various concentrations of visfatin, was found to enhance tube formation in LECs (Figure 2D). Importantly, these effects were dependent on VEGF-D, as demonstrated by the inhibition of visfatin-mediated effects when using an anti-VEGF-D antibody (Figure 2D). These findings indicate that human chondrosarcoma cells express VEGF-D in a visfatin-dependent manner, promoting lymphangiogenesis. VEGF-C is another lymphangiogenic factor; stimulation of chondrosarcoma cells with visfatin also increased VEGF-C expression (Appendix A). The role of VEGF-C in visfatin-induced lymphangiogenesis cannot be ruled out.

### 2.3. The RAF/MEK/ERK/HIF-1α Signaling Pathway Controls the Effect of Visfatin on VEGF-D Synthesis in Human Chondrosarcoma Cells

To uncover the mechanisms at play in chondrosarcoma patients, we delved into signaling pathways using data from the GSE30844 database and employed IPA software (version 23.0.; Qiagen, Hilden, Germany) for analysis. Our investigation pinpointed the RAF, MEK, ERK, and HIF-1α signaling pathways, with particular significance attributed to the HIF-1α pathway (Figure 3). We conducted experiments involving pretreatment of cells with Raf (GW5074), MEK (PD98059), and ERK (U0126) inhibitors, as well as transfection with Raf, MEK, and ERK siRNA. These interventions resulted in a inhibition of visfatin-induced VEGF-D expression, confirmed by qPCR and ELISA (Figure 4A–D). The Raf, MEK, and ERK inhibitors did not affect the basal levels of VEGF-D (Appendix A). Furthermore, visfatin promotes a time-dependent enhancement in the phosphorylation of Raf, MEK, and ERK (Figure 4E). Conversely, pretreatment with GW5074 or PD98059 markedly reduced the visfatin-induced phosphorylation of MEK or ERK (Figure 4F–G). On the other hand, treatment with a HIF-1α inhibitor (HIFi) or siRNA markedly antagonized visfatin-induced VEGF-D expression (Figure 5A–D). Visfatin stimulation increased HIF-1α expression (Figure 5E), which was inhibited by GW5074, PD98059, and U0126 (Figure 5F), indicating that visfatin enhances VEGF-D expression in human chondrosarcoma cells by activating the RAF, MEK, ERK, and HIF-1α pathways.

### 2.4. Visfatin Upregulates VEGF-D Expression by Blocking miR-2277-3p

Aberrant miRNA expression strongly influences the metastasis of chondrosarcoma [20]. Utilizing open-source software (miRWalk: http://mirwalk.umm.uni-heidelberg.de; miRDB: https://mirdb.org; miRTarBase: https://mirtarbase.cuhk.edu.cn), we identified 7 target miRNAs as potentially binding to VEGF-D mRNA (Figure 6A). Among these, visfatin primarily reduced the expression of miR-2277-3p compared to other miRNAs (Figure 6B). Furthermore, visfatin also concentration-dependently decreased miR-2277-3p expression (Figure 6C). When we transfected cells with a miR-2277-3p mimic, it downregulated the visfatin-induced increases in VEGF-D expression (Figure 6D). We then explored the relationship between miR-2277-3p and the Raf/MEK/ERK signaling pathway. Treating cells with Raf, MEK, and ERK inhibitors or their respective siRNAs antagonized the visfatin-promoted decrease in miR-2277-3p expression (Figure 6E,F). Subsequently, we investigated whether miR-2277-3p directly targets the 3′-UTR region of VEGF-D. In an examination of the VEGF-D 3′-UTR luciferase plasmids (Figure 6G), visfatin facilitated the activity of the wild-type, but not the mutant, VEGF-D 3′-UTR luciferase (Figure 6H). Transfection of cells with a miR-2277-3p mimic, Raf, MEK, and ERK inhibitors, or siRNAs reversed the visfatin-induced WT-VEGF-D 3′-UTR luciferase activity (Figure 6I–K). These findings suggest that miR-2277-3p binds to the 3′-UTR region of the human VEGF-D gene through Raf/MEK/ERK signaling following visfatin stimulation.

### 2.5. Visfatin Promotes Tumor-Related Lymphangiogenesis In Vivo

In a previous study, we reported that the overexpression of visfatin enhances chondrosarcoma-associated angiogenesis using a xenograft mouse model [20]. IHC staining revealed that overexpression of visfatin (JJ012/visfatin) led to an increase in visfatin, VEGF-D, and the lymphatic vessel marker LYVE-1 expression (Figure 7). Treatment with the visfatin inhibitor FK866 significantly suppressed the visfatin-enhanced expression of visfatin, VEGF-D, and LYVE-1 (Figure 7). Consequently, visfatin emerges as a critical factor in controlling chondrosarcoma-related lymphangiogenesis in vivo.

## 3. Discussion

Patients with chondrosarcoma are susceptible to lung metastasis, recurrence, and treatment resistance [24]. Chondrosarcoma, a heterogeneous malignant bone tumor with a high incidence of lymphangiogenesis and a propensity to spread, consists of a vital component [25]. Visfatin is closely associated with oncogenesis, and elevated levels of visfatin expression have been related to poorer prognoses in various cancer types [26,27,28,29]. In our previous studies, we discovered that visfatin promotes PDGF-C-dependent angiogenesis and metastasis in human chondrosarcoma [19,20]. Here, we also confirm that visfatin is expressed at higher levels in chondrosarcoma patients than in normal cartilage. Therefore, chondrosarcoma is a major supplier in the tumor microenvironment. Whether fibroblasts, blood, and endothelial cells also provide visfatin needs further investigation. In our current study on human chondrosarcoma, we have determined that VEGF-D is a key mediator targeted by visfatin to regulate lymphangiogenesis, both in vitro and in vivo. Visfatin enhances VEGF-D synthesis and lymphangiogenesis by activating HIF-1α and reducing miR-2277-3p expression through the Raf/MEK/ERK signaling pathway.

Lymphangiogenesis is a crucial process implicated in the dissemination of cancer cells, leading to distant metastasis that significantly influences the prognosis of patients with various cancer types [30,31,32,33,34]. This concept finds support in our study, where VEGF-D levels showed a significant correlation with the clinical stages of human chondrosarcoma. Furthermore, our animal model demonstrated that blocking visfatin using the pharmacological inhibitor FK866 effectively inhibited lymphangiogenesis. Tumor cell-driven lymphangiogenesis plays a pivotal role in lymphatic metastasis, with VEGF-D recognized as a key regulator in this process [35]. Our analyses reveal that patients with high-grade chondrosarcoma exhibit higher levels of VEGF-D expression compared to those with lower grades. Similar results were also presented in osteosarcoma, with higher VEGF-D levels than normal bone (Appendix A). Notably, visfatin induced a dose-dependent and significant increase in VEGF-D expression in chondrosarcoma cells, consequently promoting tube formation in LECs. These findings offer valuable insights into the impact of visfatin on VEGF-D-mediated lymphangiogenesis in the context of chondrosarcoma.

HIF-1α is a pivotal stress-responsive transcription factor that responds to low oxygen levels, and its expression is closely linked to tumor development and lymphangiogenesis [36,37]. In our analysis, we examined signaling pathways within the GSE30844 database using IPA software (version 23.0.; Qiagen, Hilden, Germany). Our findings indicated that the Raf, MEK, ERK, and HIF-1α signaling pathways were strongly related with the primary signaling pathway, namely, the HIF-1α pathway. Inhibition of Raf, MEK, and ERK, as well as genetic silencing using siRNAs, effectively blocked visfatin-induced VEGF-D production. Visfatin stimulation led to increased phosphorylation of Raf, MEK, and ERK, suggesting their involvement in mediating the effects of visfatin. Moreover, both the HIF-1α inhibitor and siRNA successfully mitigated visfatin-enhanced VEGF-D synthesis. In addition, the Raf, MEK, and ERK inhibitors antagonized visfatin-induced HIF-1α expression. These results collectively indicate that the Raf, MEK, ERK, and HIF-1α signaling cascades are intricately involved in visfatin-regulated VEGF-D production and lymphangiogenesis in human chondrosarcoma. To the best of our knowledge, this is the first demonstration of visfatin-HIF-1α-VEGF-D-lymphangiogenesis in chondrosarcoma. Whether the same mechanism exists in other cancer types requires further investigation.

miRNAs, which are short noncoding RNAs, play pivotal roles in various aspects of homeostasis, disease, and even the initiation of cancer [38,39,40]. Modulating miRNA expression through pharmacological interventions has the potential to effectively inhibit the migratory capacity of cancer cells, making it a promising therapeutic approach to combat tumor metastasis [41,42]. In this study, we utilized miRNA software databases (miRWalk: http://mirwalk.umm.uni-heidelberg.de; miRDB: https://mirdb.org; miRTarBase: https://mirtarbase.cuhk.edu.cn) to predict that miR-2277-3p interferes with VEGF-D transcription. miR-2277-3p was found to be involved in the dopaminergic neural pathway [43]. In addition, miR-2277-3p promotes colon cancer migration and invasion by targeting NUPR1 [44]. Our findings demonstrated that visfatin treatment led to the down-regulation of miR-2277-3p expression, and introducing the miR-2277-3p mimic into chondrosarcoma cells reversed the visfatin-induced increase in VEGF-D. Furthermore, inhibition of Raf, MEK, and ERK countered the decrease in miR-2277-3p synthesis induced by visfatin, suggesting that visfatin modulates VEGF-D-dependent lymphangiogenesis in chondrosarcoma by reducing miR-2277-3p levels through the Raf, MEK, and ERK pathway. In the VEGF-D 3′-UTR luciferase assay, it was indicated that visfatin facilitated the activity of the wild-type, but not the mutant, VEGF-D 3′-UTR luciferase. This suggests that miR-2277-3p directly binds to the 3′-UTR region of the human VEGF-D gene following visfatin stimulation. We did not examine whether HIF-1α directly binds to the VEGF-D promoter; this needs further examination.

In conclusion, our study suggests that visfatin facilitates VEGF-D-dependent lymphangiogenesis in human chondrosarcoma. Visfatin enhances VEGF-D production by activating HIF-1α and suppressing miR-2277-3p synthesis through the Raf, MEK, and ERK pathway (Figure 8). The results strongly support the notion that visfatin holds promise as a therapeutic target in metastatic chondrosarcoma.

## 4. Materials and Methods

### 4.1. Cell Line

The imortalized human lymphatic endothelial cell line, human telomerase-immortalized human dermal lymphatic endothelial cells (hTERT-HDLECs), was procured from Lonza (Walkersville, MD, USA) [45,46]. The human chondrosarcoma cell line SW1353 was bought from the American Type Culture Collection, and the JJ012 cell line was kindly provided by Dr. Sean P. Scully (University of Miami School of Medicine, Coral Gables, FL USA). Detailed information regarding its establishment and characterization is documented in previous reports [47,48]. The cells were cultured in DMEM/α-MEM medium supplemented with 10% FBS and 100 units/mL penicillin/streptomycin at 37 °C with 5% CO_2_ [49].

### 4.2. Real-Time Quantitative Polymerase Chain Reaction (RT-qPCR)

Total RNA was extracted from JJ012 and SW1353 cells using TRIzol reagent. Subsequently, reverse transcription of the messenger RNA to complementary DNA was performed using an MMLV RT kit, followed by qPCR using a Taqman assay kit. For miRNA expression analysis, qPCR was conducted using the TaqMan MicroRNA Reverse Transcription Kit and normalized to U6 expression on the StepOnePlus sequence detection system [50].

### 4.3. Enzyme-Linked Immunosorbent Assay (ELISA)

JJ012 and SW1353 cells were stimulated with visfatin, pretreated with pharmacological inhibitors for 30 min, or transfected with siRNA for 24 h. Subsequently, the cells were cultured in serum-free medium for an entire day. Following the manufacturer’s instructions, the medium was collected as conditioned medium (CM) and used to measure the expression of VEGF-D using an ELISA kit.

### 4.4. Western Blotting

For Western blotting, JJ012 and SW1353 cells (5 × 10^5^ each) were seeded in six-well plates. After 24 h of treatment with various concentrations of visfatin, the cells were collected and lysed. A total of 50 μg of protein per lane was separated on a 10% SDS-PAGE gel, transferred to a membrane, and probed with primary antibodies, including p-RAF (SC-101791, Santa Cruz Biotechnology, Santa Cruz, CA, USA), p-MEK (2338S, Cell Signaling, Danvers, MA, USA), p-ERK (SC-7383, Santa Cruz Biotechnology, Santa Cruz, CA, USA), HIF-1α (ab1, Abcam, Cambridge, UK), RAF (SC-133, Santa Cruz Biotechnology, Santa Cruz, CA, USA), MEK (SC-6250, Santa Cruz Biotechnology, Santa Cruz, CA, USA), ERK (SC-1647, Santa Cruz Biotechnology, Santa Cruz, CA, USA), VEGF-D (A19242, ABclonal, Woburn, MA, USA), and β-Actin (A5441, Sigma, St. Louis, MO, USA). Appropriate horseradish peroxidase-conjugated secondary antibodies were applied in TBS containing 5% non-fat milk. Protein bands were detected on the blots using enhanced chemiluminescence reagents (GE Healthcare Life Sciences, Glasgow, UK). The data were expressed as a relative expression ratio to β-Actin [37].

### 4.5. Tube Formation Assay

LECs were resuspended in culture media (comprising 50% EGM-2 MV BulletKit media and 50% chondrosarcoma cell-conditioned medium) and placed in 48-well plates pre-coated with 150 μL Matrigel at a density of 5 × 10^4^/100 μL. After 24 h, LECs’ tube formation was captured on camera and measured by counting the number of tube branches.

### 4.6. Immunohistochemistry (IHC) Staining

The normal cartilage and chondrosarcoma section in tissue array (OS805a) was purchased from TissueArray (Derwood, MD, USA). Human tissue sections were incubated with the primary antibodies VEGF-D (A19242, ABclonal, Woburn, MA, USA) or LYVE-1 (11-034, AngioBio Co., San Diego, CA, USA) at 4 °C overnight and subsequently incubated with a secondary antibody (1:100) for 1 h at room temperature. Finally, the sections were stained with diaminobenzidine [20].

### 4.7. Ingenuity Pathway Analysis (IPA)

A list of differentially expressed human chondrosarcoma genes from the GEO database (accession code: GSE 30844), including gene names and associated expression levels, was imported into the IPA program (version 23.0.; Qiagen, Hilden, Germany). The differentially expressed data, comprising biological processes, canonical pathways, upstream transcriptional regulators, and gene networks, were analyzed using the software’s “core analysis” tool [51].

### 4.8. Luciferase Activity

Chondrosarcoma cell lines (JJ012, SW1353) were transfected with wild-type (WT) or mutant (MUT) VEGF-D 3‘UTR luciferase plasmids using Lipofectamine 2000 and exposed to visfatin for an additional 24 h. Luciferase activity was measured using the dual luciferase assay system (Promega, Madison, WI, USA) following the manufacturer’s instructions [52].

### 4.9. Statistical Analyses

Quantified data were analyzed using GraphPad Prism 8.2 (GraphPad Software). Each experiment was repeated three times. Unless otherwise specified, data are presented as mean ± SD. Statistical significance was determined by the two-tailed unpaired Student t-test. Specific statistical analyses for each plot are described in the figure legends. A *p*-value of less than 0.05 was considered statistically significant.

## Figures and Tables

**Figure 1 ijms-25-05142-f001:**
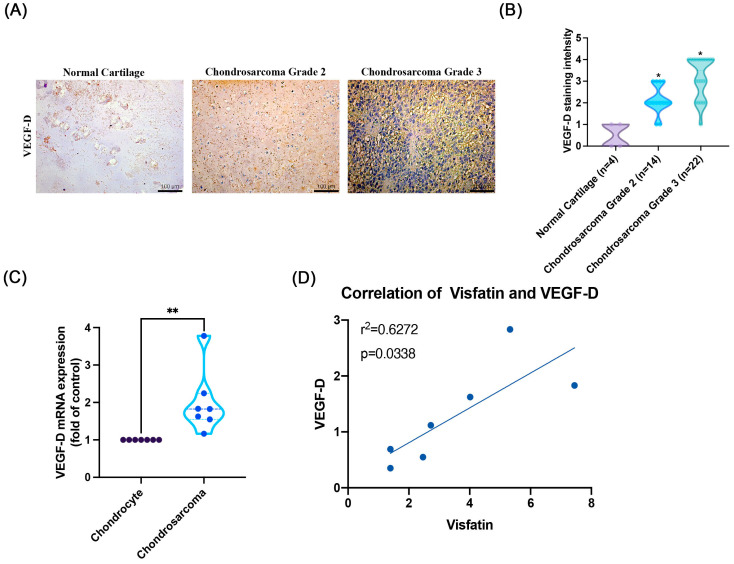
Higher levels of VEGF-D in chondrosarcoma patients. (**A**,**B**) IHC staining was performed for VEGF-D levels in chondrosarcoma patients, followed by photography and quantification. (**C**) The VEGF-D mRNA expression in chondrocytes and chondrosarcoma cells was examined by qPCR. (**D**) The levels of visfatin and VEGF-D showed a positive correlation. * *p* < 0.05 versus normal cartilage. ** *p* < 0.05 versus chondrocyte.

**Figure 2 ijms-25-05142-f002:**
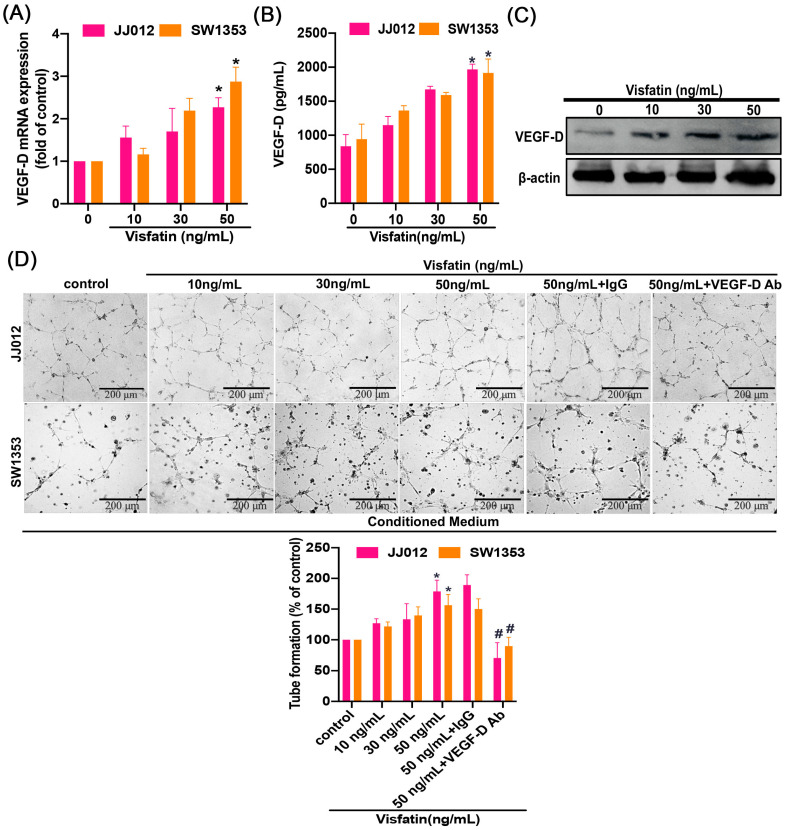
Visfatin enhances VEGF-D-dependent LEC lymphangiogenesis. (**A**,**B**) Cells were stimulated with varying concentrations of visfatin (10–50 ng/mL) for 24 h, then assayed by qPCR and ELISA for VEGF-D. (**C**) JJ012 cells were stimulated with varying concentrations of visfatin (10–50 ng/mL) for 24 h, then assayed by Western blot for VEGF-D expression. (**D**) Chondrosarcoma cells were incubated with visfatin for 24 h and then stimulated with VEGF-D or IgG antibody (1 μg/mL) for 30 min, prior to CM collection and its application to LEC. The LEC tube formation was quantified. * *p* < 0.05 versus control. # *p* < 0.05 versus visfatin-treated group.

**Figure 3 ijms-25-05142-f003:**
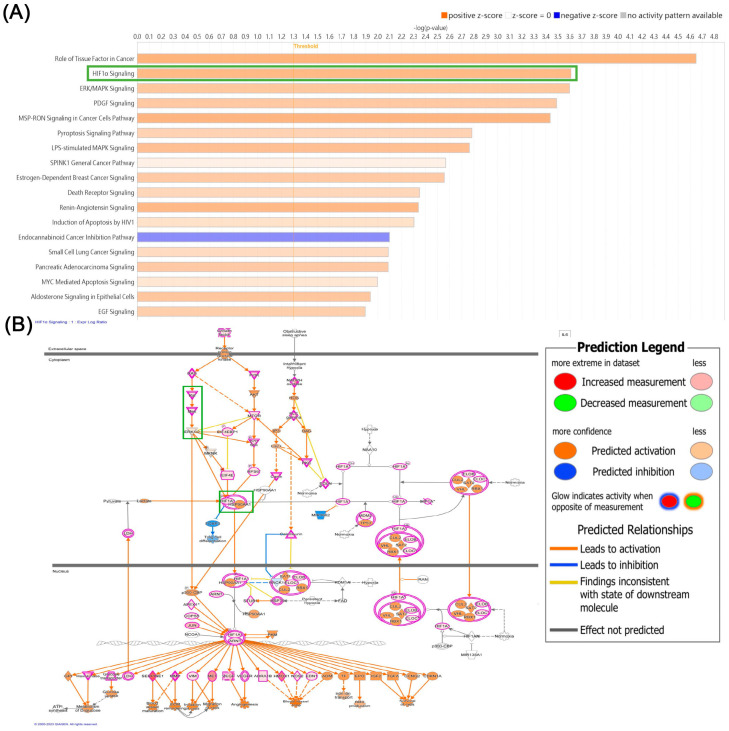
The IPA pathway enrichment in GSE30844 database. (**A**) Ingenuity Pathway Analysis pathway enrichment figure showing pathways in the GSE30844 database that were significantly elevated. (**B**) The signaling network in HIF-1α signaling. * indicated HIF1A. The green box indicated candidate signaling in current study.

**Figure 4 ijms-25-05142-f004:**
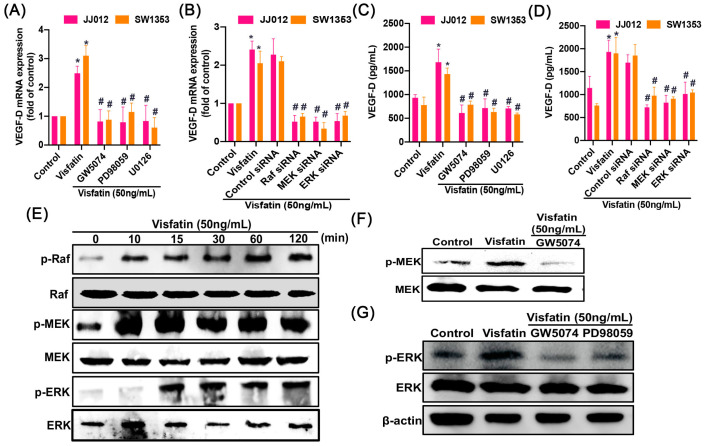
Visfatin enhances VEGF-D production in chondrosarcoma cells via the Raf, MEK, and ERK pathway. (**A**–**D**) Cells were pretreated with GW5074 (10 μM), PD98059 (10 μM), and U0126 (10 μM) for 30 min or transfected with Raf, MEK, and ERK siRNAs, then stimulated with visfatin for 24 h. The qPCR and ELISA determined VEGF-D expression. (**E**) JJ012 cells were incubated with visfatin for the indicated time intervals. The Raf, MEK, and ERK phosphorylation was measured by Western blot. (**F**,**G**) Cells were pretreated with GW5074 and PD98059 for 15 min, then stimulated with visfatin for 30 min, the MEK and ERK phosphorylation was measured by Western blot. * *p* < 0.05 versus control. # *p* < 0.05 versus visfatin-treated group.

**Figure 5 ijms-25-05142-f005:**
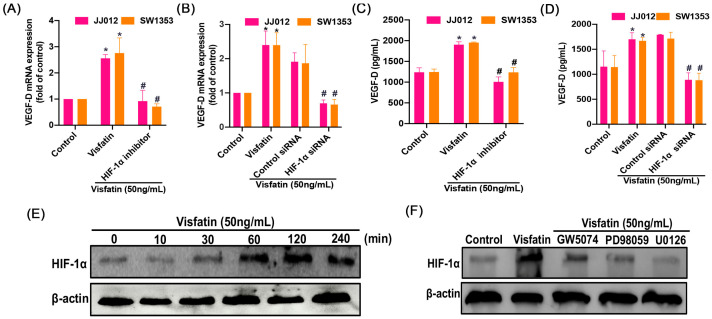
Involvement of HIF-1α in visfatin-induced VEGF-D production. (**A**–**D**) Cells were treated with an HIF-1α inhibitor (HIFi; 10 μM) for 30 min, or transfected with HIF-1α siRNA, then stimulated with visfatin for 24 h. The qPCR and ELISA determined VEGF-D expression. (**E**,**F**) JJ012 cells incubated with visfatin for the indicated time intervals or pretreated with GW5074, PD98059 and U0126 than treated with visfatin, the HIF-1α expression was examined by Western blot. * *p* < 0.05 versus control. # *p* < 0.05 versus visfatin-treated group.

**Figure 6 ijms-25-05142-f006:**
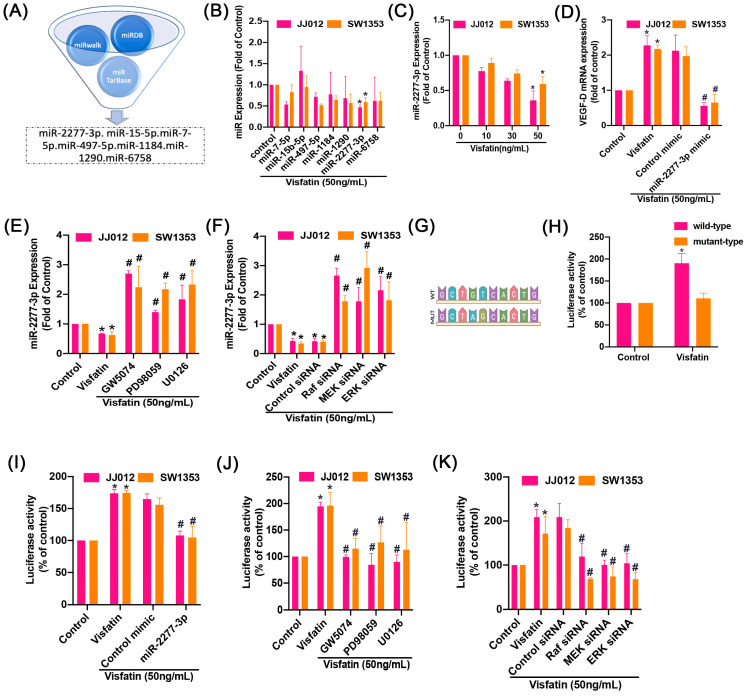
Inhibition of miR-2277-3p regulates visfatin-induced increases in VEGF-D expression of chondrosarcoma cells. (**A**) miRNA target prediction software (miRWalk: http://mirwalk.umm.uni-heidelberg.de; miRDB: https://mirdb.org; miRTarBase: https://mirtarbase.cuhk.edu.cn) identified miRNAs that potentially bind to the VEGF-D. (**B**,**C**) Cells were incubated with visfatin for 24 h, and the miRNAs expression was examined by qPCR. (**D**) Cells were transfected with miRNA mimic, then incubated with visfatin for 24 h, the VEGF-D expression was determined by qPCR. (**E**,**F**) Cells were pretreated with GW5074 and PD98059 or transfected with indicated siRNAs then stimulated with visfatin, the miRNA expression was measured by qPCR. (**G**) Schematic 3′-UTR representation of human VEGF-D containing the miR-2277-3p binding site. (**H**–**K**) Cells were pretreated with GW5074, PD98059, and U0126 or transfected with indicated siRNAs then stimulated with visfatin, the luciferase activity was measured. * *p* < 0.05 versus control. # *p* < 0.05 versus visfatin-treated group.

**Figure 7 ijms-25-05142-f007:**
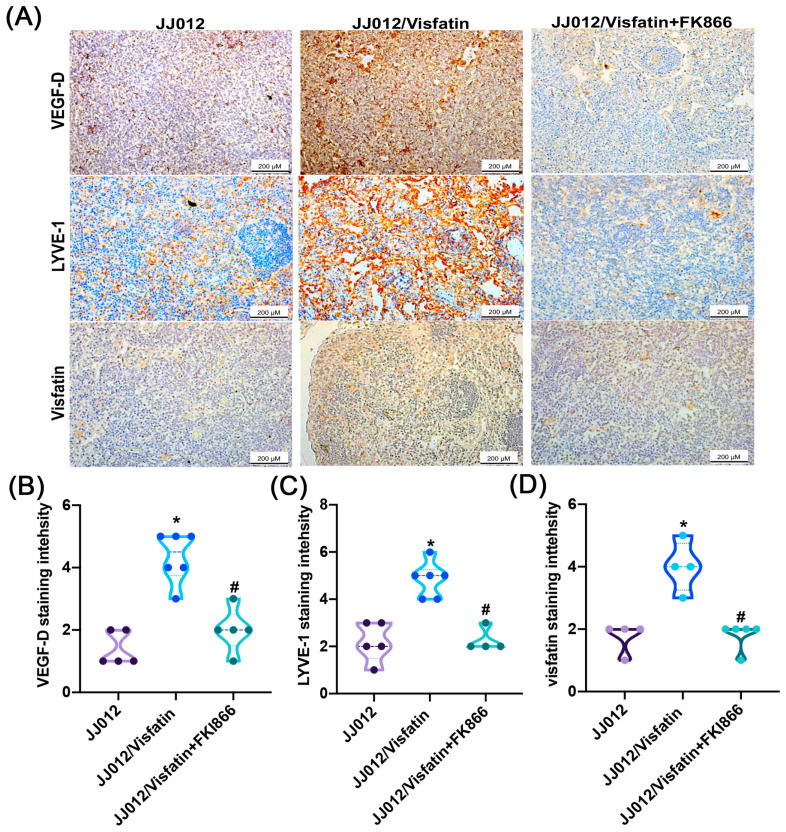
Visfatin controls chondrosarcoma-related lymphangiogenesis in vivo. IHC staining for (**A**,**B**) VEGF-D, (**A**,**C**) LYVE-1, and (**A**,**D**) visfatin in JJ012, JJ012 overexpressing visfatin (JJ012/visfatin) and JJ012/visfatin treated with FK866 (JJ012/visfatin + FK866) tumors. * *p* < 0.05 versus JJ012 group. # *p* < 0.05 versus JJ012/visfatin group.

**Figure 8 ijms-25-05142-f008:**
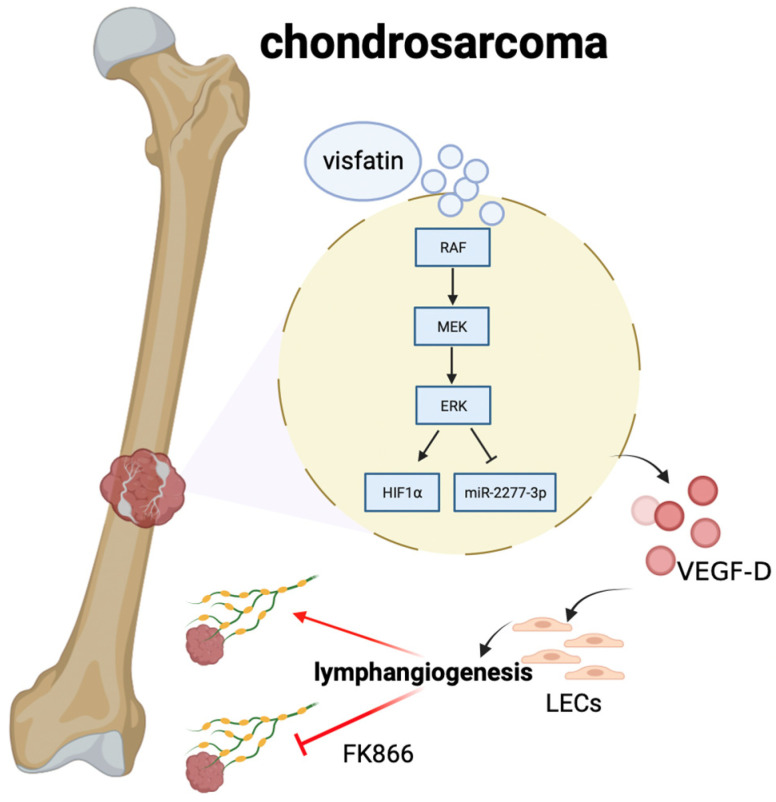
Mechanisms of visfatin-induced lymphangiogenesis in chondrosarcoma. The schematic diagram summarizes the mechanisms underlying the visfatin-induced increase in the production of VEGF-D in human chondrosarcoma and subsequent enhancement of LEC lymphangiogenesis by activating HIF-1α and inhibiting miR-2277-3p generation through Raf, MEK, and ERK signaling pathways.

## Data Availability

The original contributions presented in the study are included in the article/Appendix A, further inquiries can be directed to the corresponding author/s.

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
