# Peer review of "Visfatin Facilitates VEGF-D-Induced Lymphangiogenesis through Activating HIF-1α and Suppressing miR-2277-3p in Human Chondrosarcoma"

_ijms, 2024, doi:10.3390/ijms25105142_

Round 1
Reviewer 1 Report
Comments and Suggestions for Authors
Dear Editor,
Thank you for giving me an opportunity to review this manuscript entitled Visfatin Facilitates VEGF-D-Induced Lymphangiogenesis through Activating HIF-1α and Suppressing miR-2277-3p in Human Chondrosarcoma
This manuscript presents novelty about the conceptual flow: visfatin-HIF1a-VEGF-D-lymphangiogenesis-metastasis in chondrosarcoma
In my mind, this will be acceptable for publication if major points are addressed as below
Major points to address
1. They nicely showed visfatin-regulated lymphangiogenesis though VEGF-D in intro experiment as shown in Fig 2 A-D and in vivo in Fig 7. How about lymphangiogenesis in actual clinical samples? At the same time, visfatin staining is also recommended in clinical samples.
2. Is the pathway, visfatin-HIF1a-VEGF-D-lymphangiogenesis-metastasis, specific in chodrosarcoma? Indeed, distant metastasis such as lung or bone metastasis is sometimes observed in chondrosarcoma. But other bone sarcoma including osteosarcoma, Ewing sarcoma and so forth also shows high incidence of lung metastasis. Hence, they should investigate the expression of VEGF-D in osteosarcoma and Ewing sarcoma.
3. What cell line was used in Fig 2C?
4. As they mentioned, visfatin is one of adipokines. In chodrosarcoma, what kind of cells supplies visfatin?
Author Response
Reviewer 1
- They nicely showed visfatin-regulated lymphangiogenesis though VEGF-D in intro experiment as shown in Fig 2 A-D and in vivo in Fig 7. How about lymphangiogenesis in actual clinical samples? At the same time, visfatin staining is also recommended in clinical samples.
A: (i) We also stained for the lymphatic vessel marker (LYVE-1) and found that human chondrosarcoma expressed a higher level of LYVE-1 than normal cartilage (Supplementary Fig. 1). (Lines 90-92)
(ii) Visfatin levels are confirmed to be higher in chondrosarcoma patients than in normal cartilage (Supplementary Fig. 2). (Lines 93-95)
- Is the pathway, visfatin-HIF1a-VEGF-D-lymphangiogenesis-metastasis, specific in chodrosarcoma? Indeed, distant metastasis such as lung or bone metastasis is sometimes observed in chondrosarcoma. But other bone sarcoma including osteosarcoma, Ewing sarcoma and so forth also shows high incidence of lung metastasis. Hence, they should investigate the expression of VEGF-D in osteosarcoma and Ewing sarcoma.
A: (i) To the best of our knowledge, this is the first time demonstrating visfatin-HIF-1α-VEGF-D-lymphangiogenesis in chondrosarcoma. Whether the same mechanism exists in other cancer types requires further investigation. (Lines 259-261)
(ii) The similar results were also presented in osteosarcoma, with higher VEGF-D levels than normal bone (Supplementary Fig. 5). (Lines 240-241)
- What cell line was used in Fig 2C?
A: JJ012 cells was used in Fig 2C. (Line 126)
- As they mentioned, visfatin is one of adipokines. In chodrosarcoma, what kind of cells supplies visfatin?
A: The information has been provided. “Here, we also confirm that visfatin is expressed at higher levels in chondrosarcoma patients than in normal cartilage. Therefore, chondrosarcoma is a major supplier in tumor microenvironment. Whether fibroblasts, blood, and endothelial cells also provide visfatin needs further investigation.” (Lines 223-226)
Reviewer 2 Report
Comments and Suggestions for Authors
This research paper analyses the role of VEGF-D in human chondrosarcoma. Although of interest, several concerns exist about the basic premise's clinical proximity, and the study's significance is questionable.
#1. Who diagnosed chondrosarcoma, the subject of this study? The pathological diagnosis follows the WHO Classification as standard. Based on which grading system is Grade 4 a diagnosis?
#2. Distant metastases of chondrosarcoma are most often hematogenous, and lymphatic metastases are very rare. This study focuses on lymphangiogenesis, but is there any evidence or reports of lymphatic spread in human chondrosarcomas?
#3. There is little mention of studies using specimens from patients with chondrosarcoma, the number of samples analyzed, or essential information such as the patient profile. This means that human chondrosarcoma specimens do not support the basic premise of this study.
#4. In Figure 1C, a typographical error can be seen in the annotation. These typos suggest that this manuscript has not been thoroughly checked.
#5. The chondrosarcoma cell lines used should be annotated with the literature from which they are derived so that reference can be made to the nature of the cell lines.
#6. Ethical review approvals should be clearly stated with their approval number.
Comments on the Quality of English LanguageIt's almost good, but it should be revised for typos.
Author Response
Reviewer 2
- Who diagnosed chondrosarcoma, the subject of this study? The pathological diagnosis follows the WHO Classification as standard. Based on which grading system is Grade 4 a diagnosis?
A: (i) The normal cartilage and chondrosarcoma section in tissue array (OS805a) was purchased from TissueArray (Derwood, MD) (Lines 326-327)
(ii) Dedifferentiated chondrosarcoma is sometimes referred to as a Grade 4 lesion. We corrected the classification according to the tissue array datasheet following the WHO Classification (Fig. 1A&B).
- Distant metastases of chondrosarcoma are most often hematogenous, and lymphatic metastases are very rare. This study focuses on lymphangiogenesis, but is there any evidence or reports of lymphatic spread in human chondrosarcomas?
A: We also stained for the lymphatic vessel marker (LYVE-1) and found that human chondrosarcoma expressed a higher level of LYVE-1 than normal cartilage (Supplementary Fig. 1). (Lines 90-92)
- There is little mention of studies using specimens from patients with chondrosarcoma, the number of samples analyzed, or essential information such as the patient profile. This means that human chondrosarcoma specimens do not support the basic premise of this study.
A: The normal cartilage and chondrosarcoma section in tissue array (OS805a) was purchased from TissueArray (Derwood, MD) (Lines 326-327). The number of samples has been provided in Fig. 1B.
- In Figure 1C, a typographical error can be seen in the annotation. These typos suggest that this manuscript has not been thoroughly checked.
- The typographical error has been corrected in Fig. 1C.
- The chondrosarcoma cell lines used should be annotated with the literature from which they are derived so that reference can be made to the nature of the cell lines.
A: The information has been provided. “The human chondrosarcoma cell line SW1353 was bought from the American Type Culture Collection, and the JJ012 cell line was kindly provided by Dr. Sean P. Scully (University of Miami School of Medicine, USA)” (Lines 289-291)
- Ethical review approvals should be clearly stated with their approval number.
A: The information has been provided. (Lines 357-361)
Reviewer 3 Report
Comments and Suggestions for Authors
In the present study, the authors reported the levels of VEGF-D related with the development of human chondrosarcoma and the adipokine Visfatin could facilitate this regulation by Raf/MEK/ERK signaling pathway. The finding is interesting and meaningful, and the quality of data is decent. But there are some questions listed need to be figured out before publication.
1. The authors have published another paper that reported Visfatin regulated VEGF-C also by MEK/ERK pathway, will VEGF-C also be regulated in chondrosarcoma models or VEGF-D also be regulated in ESCC models? Why is the difference if VEGF-C and VEGF-D are regulated differently in these two diseases?
2. What are the mechanisms underlying the regulation of VEGF-D by HIF-1a and miR-2277-3P? Are there any targets of miR-2277-3P have been reported?
3. Have you ever tried to treat Raf, MEK, ERK inhibitors or siRNAs directly (without Visfatin) to see their effects on VEGF-D levels? If Raf, MEK, ERK inhibition can affect VEGF-D levels, it is not enough to conclude that Visfatin affects VEGF-D levels by this pathway.
4. Line 101, there is a “VEGF-C protein” should be VEGF-D.
Comments on the Quality of English LanguageNo comments.
Author Response
Reviewer 3
- The authors have published another paper that reported Visfatin regulated VEGF-C also by MEK/ERK pathway, will VEGF-C also be regulated in chondrosarcoma models or VEGF-D also be regulated in ESCC models? Why is the difference if VEGF-C and VEGF-D are regulated differently in these two diseases?
A: (i) VEGF-C is another lymphangiogenic factor; stimulation of chondrosarcoma cells with visfatin also increased VEGF-C expression (Supplementary Fig. 3). (Lines 120-121)
(2) The information has been mentioned. “The role of VEGF-C in visfatin-induced lymphangiogenesis cannot be ruled out.” (Lines 121-122)
- What are the mechanisms underlying the regulation of VEGF-D by HIF-1a and miR-2277-3P? Are there any targets of miR-2277-3P have been reported?
A: (i) The information has been discussed. “In the VEGF-D 3'-UTR luciferase assay, it was indicated that visfatin facilitated the activity of the wild-type, but not the mutant, VEGF-D 3'-UTR luciferase. This suggests that miR-2277-3p directly binds to the 3'-UTR region of the human VEGF-D gene following visfatin stimulation. We did not examine whether HIF-1α directly binds to the VEGF-D promoter; this needs further examination.” (Lines 275-279)
(ii) The information has been added. “miR-2277-3p was found to be involved in the dopaminergic neural pathway [43]. In addition, miR-2277-3p promotes colon cancer migration and invasion by targeting NUPR1 [44].” (Lines 267-269)
- Have you ever tried to treat Raf, MEK, ERK inhibitors or siRNAs directly (without Visfatin) to see their effects on VEGF-D levels? If Raf, MEK, ERK inhibition can affect VEGF-D levels, it is not enough to conclude that Visfatin affects VEGF-D levels by this pathway.
A: The result has been provided. “The Raf, MEK and ERK inhibitors didn’t affect the basal levels of VEGF-D (Supplementary Fig. 4).” (Lines 140-141)
- Line 101, there is a “VEGF-C protein” should be VEGF-D.
A: The “VEGF-C” has been corrected to “VEGF-D” (Line 110)
Round 2
Reviewer 1 Report
Comments and Suggestions for Authors
They completely addressed my suggested points.
Author Response
Reviewer 1
They completely addressed my suggested points.
A: We appreciate the reviewer's comment.
Reviewer 2 Report
Comments and Suggestions for Authors
The authors have mostly responded to my requests, but one point is that for the cell lines used, they should not only describe how you obtained them, but they should also cite references that describe the characteristics of the cells. It would be preferable to cite the report on the establishment of the cell line or the experimental paper in which the cell line was first used.
Comments on the Quality of English LanguageThe English language description provided is generally acceptable.
Author Response
Reviewer 2
The authors have mostly responded to my requests, but one point is that for the cell lines used, they should not only describe how you obtained them, but they should also cite references that describe the characteristics of the cells. It would be preferable to cite the report on the establishment of the cell line or the experimental paper in which the cell line was first used.
A: The more detail information has been provided. “Detailed information regarding its establishment and characterization is documented in previous reports [47, 48].” (Lines 290-292).
Reviewer 3 Report
Comments and Suggestions for Authors
All the questions are addressed, and no more comments.
Author Response
Reviewer 3
All the questions are addressed, and no more comments.
A: We appreciate the reviewer's comment.